# Biogeographical and Ecological Patterns of the Bryophytic Flora Inhabiting the Small Islands Surrounding the Italian Peninsula, Sicily and Sardinia

**DOI:** 10.3390/plants14111618

**Published:** 2025-05-26

**Authors:** Silvia Poponessi, Michele Aleffi, Annalena Cogoni, Antonio De Agostini

**Affiliations:** 1Institute for Alpine Environment, Eurac Research, via Druso 1, 39100 Bozen, Italy; 2School of Biosciences and Veterinary Medicine, University of Camerino, via Pontoni 5, 62032 Camerino, Italy; michele.aleffi@unicam.it; 3Department of Environmental and Life Science, Botany Section, University of Cagliari, Viale S. Ignazio 13, 09123 Cagliari, Italy; cogoni@unica.it (A.C.); antonio.dea@unica.it (A.D.A.); 4Co.S.Me.Se., Consorzio per lo Studio dei Metaboliti Secondari, 09132 Cagliari, Italy

**Keywords:** Mediterranean islands, bryophytes, ecology, stress resistance

## Abstract

Bryophytes’ adaptability and stress resistance make them excellent colonizers. Moreover, bryophytes are key components of almost all terrestrial ecosystems from aquatic to arid to freezing cold. Bryophytes are also unique models to study adaptation and stress resistance in plants. Bryological studies in the Mediterranean area are mainly floristic-oriented, and consequently, the knowledge of the autoecology of the species inhabiting Mediterranean islands and islets is very scarce. The aim of this study is to evaluate bryophyte diversity in a number of islands and islets surrounding the Italian peninsula, Sicily and Sardinia. Moreover, based on the geographical and environmental features available for the studied islands, we analyze the role of different ecological variables (such as the islands’ area, insularity degree, altitude, and substratum type) in shaping bryophytic species richness and diversity. In the present study, ecological indicators adapted to Mediterranean bryophytes were also used to describe from an ecological and functional viewpoint the species inhabiting the studied islands and to explore to what extent the islands’ features have had a role in shaping the ecological features of the bryophytic floras inhabiting them. Within this study, an updated overview on the floristic richness and diversity of the small islands surrounding the Italian peninsula, Sicily and Sardinia was presented. The subject of the discussion was the island-related floristic and ecological differences as well as the drivers of these differences.

## 1. Introduction

Islands possess distinctive ecological and botanical profiles characterized by unique strengths and vulnerabilities. Their geographical isolation from the influences of the mainland (degree of insularity, here intended as the distance from the nearest mainland or major island) often results in a high degree of presence of endemisms, making them rich in peculiar species and ecosystems. Nevertheless, the aforementioned isolation can also render island ecosystems, which are particularly fragile, highly susceptible to external threats. It is notable that islands frequently support a diversity of unique species that are not found on the mainland, and they often have a higher proportion of endemic species than continental areas [1]. This phenomenon facilitates the establishment of distinctive ecosystems that contribute substantially to global biodiversity [2]. However, the insularity of these ecosystems often results in a paucity of evolved defenses against invasive species, which can outcompete native species, leading to extinctions. This phenomenon constitutes a primary driver of biodiversity loss on islands [3]. Bryophytes are of considerable scientific importance in island environments, where they offer crucial ecological services. Furthermore, bryophytes offer fundamental insights into the field of evolutionary biology and function as indicators in ecological monitoring and conservation efforts. The preservation of these keystone species and their ecosystems across the world’s islands is contingent on the implementation of dedicated research and conservation strategies [4]. Although bryophytes are often considered a neglected and understudied plant group, the late 19th century saw an increase in the study of Italian bryophytes. This was due to the notable contributions of botanists who conducted extensive fieldwork, documenting species across various Italian regions, including the islands of Sardinia and Sicily [5,6]. The mid-20th century witnessed the advent of a more systematic approach to bryological research in Italy, which was marked by the inception of ecological and phytogeographical studies. Indeed, researchers have begun to recognize the significance of bryophytes in assessing ecological health, especially in fragile Mediterranean ecosystems [7]. Noteworthy studies by botanists in the 1950s began to emphasize the biogeographical patterns of bryophytes in the rich Italian ecosystems, including those of Sicily and Sardinia. Giacomini’s contributions, including research on the Mediterranean mosses, established the foundation for future ecological assessments in a context increasingly challenged by human impact and climate change [8]. In recent years, a considerable amount of new research has been conducted on the bryological flora of Mediterranean major and smaller islands [9,10,11,12,13,14,15,16,17].

In light of the aforementioned points, the present study aims to contribute to the expansion of knowledge on the subject of bryophyte diversity and ecology in the distinctive context of Mediterranean islands and islets. To this end, it will address the bryological diversity of the islands of the islands and islets surrounding the Italian peninsula, Sicily and Sardinia (the two largest islands of the Mediterranean basin). The subsequent stage of the research program involved the evaluation of bryological diversity across the selected islands. This was followed by an exploration of the correlation between this diversity and the geographical and environmental factors that were identified as having an impact on the sites where the bryological flora was present. The characterization of the bryological floras of the studied islands is finally undertaken in terms of the autoecology of the species, which is defined as the species’ preferences for environmental features such as light radiation, humidity and soil fertility. In order to accomplish this objective, it was necessary to construct a database that incorporated species data (presence/absence) derived from previous scientific studies and personal sampling campaigns. This preliminary study provides a concise overview of the bryoflora of the Italian islands and islets, offering preliminary insights into the diversity of these non-vascular plants within the insular framework. The study also includes an analysis of the role of geographical and environmental factors in shaping such diversity.

## 2. Results

The present study examines the bryophytes forming part of the floras of the islands and islets surrounding the Italian peninsula, Sardinia and Sicily (Figure 1). The islands studied were very different from a geographical point of view (the main characteristics of the islands and islets belonging to the archipelagos studied are presented in Appendix A).

The process of the database construction and pruning, as described in the Materials and Methods section, resulted in the inclusion of 365 species proceeding from 60 islands and islets so distributed in the three archipelagos: 18 circum-Italian, 17 circum-Sardinian and 25 circum-Sicilian. The database entries considered in the present study belong to each of the three divisions of bryophytes (hornworts, liverworts and mosses), and the species included in the study are divided into 67 families and 147 genera. The way in which species are shared between or exclusive to certain archipelagos is show in the Venn diagram in Figure 2.

The diagram shows that a core of 95 species is common to the three archipelagos. While 95 species are found only in the circum-Italian archipelago, 50 species are found only in the circum-Sicilian archipelago and 13 species are found only in the circum-Sardinian islands and islets. As might be expected, the number of species shared by the circum-Italian and the circum-Sicilian archipelagos is higher (85) than the number of species shared by the circum-Sardinian archipelago and the circum-Italian and the circum-Sicilian archipelagos (respectively, 17 and 10). In particular, the recent sampling campaigns focused on the Sardinian area have allowed the identification of new species for the bryological flora of Sardinia. According to [18], we report the following: *Syntrichia sinensis* (Müll.Hal.) Ochyra found on Tavolara; *Sciuro-hypnum plumosum* (Hedw.) Ignatov & Huttunen foundon the island of S. Stefano (La Maddalena); *Campylopus subulatus* Schimp. ex Milde found in the island of Carloforte; and finally *Tortella fragilis* (Drumm.) Limpr found in the island of Spargi.

In order to understand the relationships between geographical features and the characteristics of the floras, it is necessary to study both simple taxonomic richness and ecological diversity. In order to have a quick representation of both in the studied archipelagos, the species richness, the genera-to-species ratio (G.S.R. from now on), the families-to-genera ratio (F.G.R. from now on) and the area-corrected Simpson index were calculated and compared (Figure 3). The species richness and G.S.R. significantly differed in the archipelagos, the species richness being higher in the circum-Italian archipelago while the G.S.R. being higher in circum-Sardinian islands. F.G.R. is uniform in the studied archipelago, and so is the area-corrected Simpson index.

To explore the extent to which the geographical features of the studied islands may have influenced bryological diversity (considering together the three archipelagos), the correlation of G.S.R., F.G.R and the area-corrected Simpson index with the islands’ geographical features were explored and represented by bar graphics (Figure 4). The analysis revealed that the three indexes considered decreased in larger islands. G.S.R. and F.G.R. decreased eastward (negative correlation with longitude), while G.S.R. increased northward. Finally, the area-corrected Simpson index tends to decrease in islands presenting higher maximum altitude. However, these correlations should be interpreted as trends emerging considering islands apart, so that does not necessarily reflect in significant differences in the indexes when considering the studied islands on their whole (as reported in Figure 3).

From an ecological point of view, the species inhabiting the archipelagos presented a diverse autoecology, intended as the requirements and preferences in terms of soil pH, water availability, temperature and light as described by the indicators of choice which are the Dierßen indicators adapted for bryophytes [19] (Figure 5). Of the four indicators considered, only temperature and light, which describe preferences in terms of ambient temperature and solar radiation, respectively, differed significantly among the three archipelagos. Specifically, with regard to temperature, the values measured in the circum-Sicilian islands were higher than those measured in the circum-Italian islands, while the preferences with regard to light described more heliophilic species in the circum-Sardinian islands than in the circum-Italian islands.

In order to ascertain the extent to which geographical features of the colonized islands may have selected for certain ecological preferences in the bryophytes inhabiting them, correlations are drawn in Figure 6. The following graphic illustrates the correlations between the geographical features of the islands and the autecology of the species in question in terms of soil acidity, environmental humidity, temperature and light preferences. A number of significant correlations were identified between bryophytes, their autecology and the geographical features of the islands and islets they colonize. In particular, the findings of the present study demonstrate a southward trend in species preferences, with an increasing propensity for environments that are more illuminated and warmer, which is concomitant with a decline in humidity requirements. In addition, species exhibiting a southward tendency demonstrated a propensity for higher soil pH levels. With regard to longitude, bryophytes and floras are more termophilous in the east. In conclusion, a negative correlation between island area and light requirements was observed. This suggests that smaller islands generally support bryophyte floras that are more tolerant of higher solar radiation. In conclusion, with a focus on island maximum altitude, it was determined that higher islands are colonized by a greater number of acidophilous and sciaphilous bryophyte species.

## 3. Materials and Methods

### 3.1. Geology of Study Sites

The Italian archipelago, comprising numerous islands such as those in the Tyrrhenian Sea, including the Aeolian, Egadi, and Sardinian islands, is predominantly shaped by the ongoing collision between the African and Eurasian tectonic plates. This tectonic interaction has been a catalyst for significant geological activity, including orogeny (mountain building) and volcanism, both of which are evident throughout the islands. The interaction of these tectonic forces has resulted in the formation of numerous faults and folds, which are prominent features in the geology of central and southern Italy [20]. The islands of Sicily and Sardinia, specifically, have been influenced by the complex tectonic setting of the Mediterranean region, which has shifted over geological time scales due to plate dynamics (Figure 1). Volcanism has been identified as a significant geologic phenomenon in the history of numerous Italian islands, including the Aeolian Islands, the Campanian Archipelago, and the Pontine Islands, and, to a lesser extent, the Tuscan Archipelago. It is evident that this geologic feature has exerted a substantial influence on the distribution and diversity of bryophyte species within these environments. The Aeolian Islands, in particular, are predominantly volcanic in origin, featuring active volcanoes such as Stromboli. This has provided geologists with a valuable opportunity to study volcanic activity and evolution first-hand [20]. The islands have also undergone significant erosional and sedimentary processes shaped by climatic factors and sea-level changes. The interaction between these processes has resulted in the formation of distinct morphological features, including cliffs, coves, and sandy beaches. The Sardinian archipelago’s geology is indicative of its complex tectonic history, as evidenced by the presence of ancient crystalline rocks, sedimentary basins, and areas of volcanic activity. The island’s geology is characterized by Paleozoic formations, predominantly comprising granites and schists, which were formed during the Variscan orogeny [21].

### 3.2. Data and Statistical Analysis

The present investigation is predicated upon a dataset that has been constructed through the interaction of data from past scientific literature, from the Italian bryophytes database (from which the species associated with each island forming part of the studied archipelagos were extracted), and by the identification of samples collected in several personal sampling campaigns completed between 2020 and 2023 in the Sardinia region. The issue we encountered during the setup of this study was the unknown and uneven sampling effort behind each species list and bryoflora in our possession, derived from data of such variegated origin. To avoid including species lists resulting from inadequate sampling efforts, we applied a threshold to exclude such records from the analysis. This threshold was based on the species-to-area ratio (hereafter S/A ratio), calculated as the number of species reported for a given island divided by its area in square kilometers. To be precise, species lists featuring an S/A ratio less than 2 were eliminated from the study. The chosen threshold should not be intended as a way of modeling the species-to-area relationship in a strict ecological sense, given the recognition that the species–area relationship is generally non-linear. Nonetheless, in circumstances where there is an absence of standardized sampling effort data, a threshold based on the species-to-area ratio constitutes a pragmatic and replicable methodology for the retention of as much pertinent information as possible whilst concomitantly minimizing the incorporation of under sampled islands. It is important to note that the aforementioned threshold was utilized as a rudimentary and prudent filter to discern islands whose diminished species counts might be indicative of inadequate sampling efforts. This methodological approach enabled the preservation of valid and potentially valuable data, particularly from smaller islands where comprehensive surveys may not exist, while concomitantly excluding extreme outliers with the potential to bias subsequent diversity analyses. Furthermore, the S/A ratio has been utilized as an index in previous studies that addressed the bryophytes diversity at the larger scale, as evidenced in [18]. In the initial phase of the study, 92 islands were included in the analysis. However, following a rigorous process of data refinement, the number of islands considered in the analysis was reduced to 60. The islands selected for inclusion in the study were categorized into three distinct archipelagos: the circum-Italian archipelago, encompassing the islands and islets surrounding the Italian peninsula, comprising a total of 18 islands; the circum-Sicilian archipelago, consisting of 25 islands; and the circum-Sardinian archipelago, which includes 17 islands. The nomenclature of species and the distribution of bryophytes are in accordance with [18]. The species lists for each archipelago were used to create the Venn diagram in Figure 2. In the present study, three indicators were included to evaluate and compare bryological biodiversity: species richness, the genera-to-species ratio (G.S.R.), the families-to-genera ratio (F.G.R.), and the Simpson diversity index calculated based on presence/absence data and corrected for island area, due to a statistically strong correlation between the index and island size. Indeed, although the Simpson index is conventionally calculated using species abundance to reflect both richness and evenness, it can also be adapted for presence/absence data. In this particular context, the index is reduced to a function of species richness, as all species are given equal weighting. In order to account for the confounding effect of island area on biodiversity, a linear regression model was applied with the Simpson index as the dependent variable and the natural logarithm of island area as the independent variable. Given the model demonstrated a significant positive relationship between island area and the Simpson index, the residuals of this regression were utilized as an area-corrected diversity metric. In instances where data pertain to archipelagos, such as in the case of the number of taxa, biodiversity indexes, geographical features and ecological indicators, these values are obtained by averaging the values of the features of the islands or of the species forming part of the archipelagos. The present study sought to ascertain the correlation between a number of geographical features of the islands and the associated bryological floras. The geographical factors that were included and analyzed for each island are the following: mean longitude, mean latitude, area (km^2^), degree of insularity (intended as the distance separating the island from the nearest mainland or major island (m), and the maximum altitude measured in meters above sea level (a.s.l.). In relation to the ecological indicators of the species under study, the requirements in terms of acidity, humidity, temperature and light were associated to each species in accordance with the traits proposed by [19]. These traits are well suited to reflect the species-specific environmental requirements and preferences of the species in the study area, which is characterized by a purely Mediterranean climate. The statistical analysis comprised an assessment of the statistical significance of the differences in the indexes describing bryological diversity and ecological requirements among the three archipelagos conducted by means of a parametric ANOVA. The results of the post hoc tests results were reported as a compact letter display above the boxplots (differences were considered to be significant at *p*-values less than 0.05). The correlation among the geographical features of the islands forming part of the archipelagos with species richness, F.G.R., G.S.R. and the area-corrected Simpson index, as well as with the species’ ecological indicators, were explored and represented by bar graphics. The implementation of the Spearman method was utilized for the purpose of testing the correlations. The statistical significance of the observed correlations was determined by applying a *p*-values cut-off of 0.05, which is a commonly accepted threshold for statistical significance in scientific analysis. All statistical analyses were carried out using R-software, version 4.1.3 (10 March 2022) [22] implemented with the following packages: “ggpubr” and “dplyr” for boxplots, “venn” for the Venn diagram, and “lares” for the representation of correlations.

## 4. Discussion

The Mediterranean basin is widely acknowledged as one of the world’s biodiversity hotspots with profound ecological and evolutionary significance. The distinctive characteristics of this region are attributable to its unique climate, which is characterized by hot, dry summers and mild, wet winters, as well as a rich variety of habitats. In this context, bryophytes (mosses and liverworts) play a key role, also serving as bioindicators of habitat health due to their sensitivity to environmental changes. However, the bryophyte flora inhabiting Mediterranean islands is also particularly vulnerable to the impacts of contemporary climate change, including sea-level rise, temperature shifts, altered precipitation patterns and increased storm intensity. In island contexts, the small size and isolation of islands are well-documented factors that further increase the vulnerability of bryophytes to extinction. In this study, an analysis of bryophyte biodiversity patterns across Mediterranean islands and islets was conducted with the aim of exploring the relationships between the geographic features of the studied islands and the diversity and ecological preferences of the bryofloras inhabiting them. The initial approach to species distribution, as illustrated in the Venn diagram in Figure 2, revealed a core of 95 species shared among the three archipelagos under study along with significant overlap between the circum-Sicilian and circum-Italian bryological floras. With regard to the species described in the circum-Italian and circum-Sicilian islands, the higher number compared to the species described in the circum-Sardinian islands should be interpreted considering the historically lower sampling effort directed toward the bryoflora of the islands surrounding Sardinia. Consequently, a lower number of records are attributable to the present study. Figure 2 is also informative regarding the extent to which bryophytes are able to disperse and distribute across longer distances in the Mediterranean basin considering the low level of species overlap between the circum-Sardinian archipelago and the circum-Italian and circum-Sicilian ones. It is imperative to interpret these results in light of the negligible dispersal of Mediterranean bryophytes by anemochoria. Mediterranean bryophytes have indeed been observed to implement a “annual shuttle species” strategy, which is characterized by the prioritization of vegetative reproduction over sexual reproduction [23]. The outcome of this process is the production of propagules and/or persistent protonema, which are essential for ensuring vegetative reproduction. This is achieved by means of spores of a greater length, shorter setae and a capsule that is immersed in the gametophyte body. This, in turn, results in the dispersion of spores over great distances being less likely. The findings presented in Figure 2 concerning the imbalance of species observed in the circum-Sardinian islands in relation to the remaining archipelago are corroborated, at least in part, by the interpretation of Figure 3. In fact, while the circum-Italian archipelago is the one featured by the higher species richness (especially with respect to the circum-Sardinian islands), when addressing the richness in terms of genera and families, the circum-Sardinian islands tend to present higher values. This suggests that Sardinia and its surrounding islands may be a promising, yet underexplored, bryophyte biodiversity hotspot. In terms of the relationships between the geographical features of the studied islands and bryophyte diversity, it was reported here that greater islands exhibited a decrease in all three indexes under consideration. This is at odds with the conventional island biogeography theory [24]. It is thought that this may conventionally predict an increase in species richness and diversity on larger islands due to reduced extinction rates and enhanced habitat heterogeneity. It is recommended that the discrepancy identified in the present study should be interpreted in light of the characteristics of the data utilized, which, even after meticulous filtering to minimize the impact of underexplored islands (as outlined in the Materials and Methods section), may still be influenced by such instances. Bryological explorations, particularly in the Mediterranean islands and islets, are characterized by a considerable degree of fragmentation and frequently fall short of the requisite level of sampling effort. It is possible that the conservative approach employed in this study to prevent the loss of valuable information regarding bryophyte diversity may have resulted in the inclusion of underexplored islands. Consequently, this may have led to bryophyte diversity relationships that are incongruent with the prevailing principles and trends of island biogeography. In addition to the aforementioned observations, a decline in both the G.S.R. and F.G.R. indexes was evident with increasing longitude (eastward), thereby validating the findings presented in Figure 3. It is evident that the region in question exhibits a promising level of bryophyte diversity, extending to the surrounding Sardinian islands and islets. Another emerging trend is the decrease in the area-corrected Simpson index in islands where the maximum elevation is greater. This finding indicates that elevational complexity alone does not necessarily enhance ecological evenness when island size is considered. The hypothesis that higher altitudes may impose environmental stressors (e.g., harsher microclimates, limited niche partitioning) that reduce competitive exclusion is presented. This is postulated to result in the favoring of dominant species over a balanced community. The absence of significant variation among the biodiversity indexes included in the study in relation to insularity may be attributable to the negligible migration capability of bryophytes in the Mediterranean region. Consequently, it is reasonable to hypothesize that the migration of bryophytes between neighboring islands is minimal. In this context, the function of intermediate stations in facilitating migration is of paramount importance. According to [25], the impact of distance can be mitigated by the presence of ‘relay islands’ (‘stepping stone islands’) located at intermediate distances. These islands serve to reduce the effective isolation between islands and continents. In addition to the previously discussed factors influencing the distribution of flora across islands, including geographical and environmental factors such as barriers between islands and continental populations, prevailing winds, and sea currents, it is imperative to acknowledge the impact of human activity on biodiversity. The ongoing and significant threat posed by human activities to the biodiversity of islands necessitates further consideration in order to ensure the preservation of island ecosystems. A variety of factors have been identified as having a considerable impact on the biodiversity of the species. These include agro-silvo-pastoral activities, the percentage of cultivated areas, the number of domestic animals, the percentage of forest cover, residential settlements, hotel facilities, and the seasonal influx of tourists to the most accessible islands by boat, ferry, etc. Moreover, in the case of volcanic islands, the nature of the underlying rock significantly impacts the development of endemic species. In addition to the edaphic characteristics, the presence of hot springs and effusive phenomena such as fumaroles, calderas, and in general phenomena of secondary volcanism with the emanation of steam and other volcanic gases, which are generally present in the vicinity of craters, is also decisive. For instance, *Rhynchostegium strongylense*, which was documented as the unique record for Europe by [25,26] for the island of Stromboli (sub. *Barbella strongylensis* Bott.), is a prime example. It was subsequently discovered by [26] on the island of Vulcano, by [25] on the island of Pantelleria, and by [27] on the island of Ischia. The species under discussion is closely linked to the presence of fumarolic stations and was an Italian endemism for many years. Another species associated with hot–humid environments at the mouth of caves with fumaroles is *Calymperes erosum*, which was collected by Sommier in 1906 on the island of Pantelleria and described by Bottini as *Calymperes sommieri* Bott. This species was also considered to be an Italian endemic for a considerable period. However, a subsequent revision by [28] resulted in its synonymizing with *C. erosum*, which is a tropical species widespread in South America. Notwithstanding, the presence of this species on the island of Pantelleria is of particular interest, as it is the only species of the genus *Calymperes* in Europe. A third example is *Trematodon longicollis* Michx., a sub-tropical species, reported in Europe only for the island of Crete and present in Italy near the fumaroles of Pantelleria, Ischia, Vesuvius and the Solfatara of Pozzuoli. All these species are considered threatened due to the vulnerability of the habitat at both Italian and European levels [23,29]. In the present study, the focus was also directed toward the autoecology of bryophytes inhabiting Mediterranean islands and islets. As illustrated in Figure 5, the initial observation indicates a uniformity in the requirements concerning soil pH and water availability. However, the temperature and light preferences exhibited variability across the archipelagos under study. As would be anticipated, bryophytes on circum-Sicilian and circum-Sardinian islands and islets demonstrate a higher degree of tolerance to elevated temperatures and solar radiation in comparison to their circum-Italian counterparts. Sardinia and Sicily are subject to the typical Mediterranean climate, characterized by warm temperatures and several bright and clear days per year, due to their latitude. In order to explore this topic further, the correlation in Figure 6 was investigated with the aim of understanding the extent to which the geographical features of the islands may have determined specific ecological preferences in the bryophytes colonizing them. The observations made in this study partially corroborated the findings of the preceding analysis. It was evident that there was an increase in temperature and light requirements in the southern region, which was concomitant with a decrease in humidity requirements. This finding lends further support to the hypothesis that in response to a shift toward a more Mediterranean climate, the bryological flora undergoes adaptation, manifesting species with enhanced tolerance to arid, hot and high–light environments. Furthermore, this phenomenon was observed to occur in an eastward direction, where bryophytes exhibited greater thermophilous characteristics. This phenomenon is likely attributable to the influence of circum-Sicilian floras. The analysis of correlations demonstrated that bryophytes inhabiting smaller islands exhibited increased tolerance to enhanced solar radiation. In contrast, more sciaphylous species were present in islands characterized by higher maximum altitude. The smaller islands included in the study are indeed minuscule, and in such contexts, the extent of vegetation is negligible, not to mention the absence of tree canopy, which should be considered wholly absent. It is important to note that the screening effect offered by superior plants should be excluded in order to ascertain the extent to which bryophytes colonizing smaller islands are affected by extreme solar radiation. Conversely, islands with higher maximum altitudes exhibit greater complexity in terms of vegetation and topography, resulting in the creation of numerous micro-niches that are conducive to the colonization by sun-avoiding species, thereby ensuring their protection from solar radiation.

## 5. Conclusions

In the present study, given the limitations linked to heterogeneous data proceedings, new knowledge was gained about the features of bryophyte floras in the Mediterranean islands context. In a scenario of climate change, islands are particularly vulnerable in terms of loss of biodiversity, and this is particularly true for byrophytes. In this sense, studies like the present one may help achieve the effective conservation of these key components of terrestrial ecosystems, helping build informed policy frameworks. It is clear that further exploration of the less studied islands is imperative to ensure more homogeneous data collection in the future years which, in turn, will grant more informative analysis and sound knowledge. This study is conceived as a preliminary but crucial effort to elucidate the current state of knowledge on the bryofloras featuring the Italian main archipelagos.

## Figures and Tables

**Figure 1 plants-14-01618-f001:**
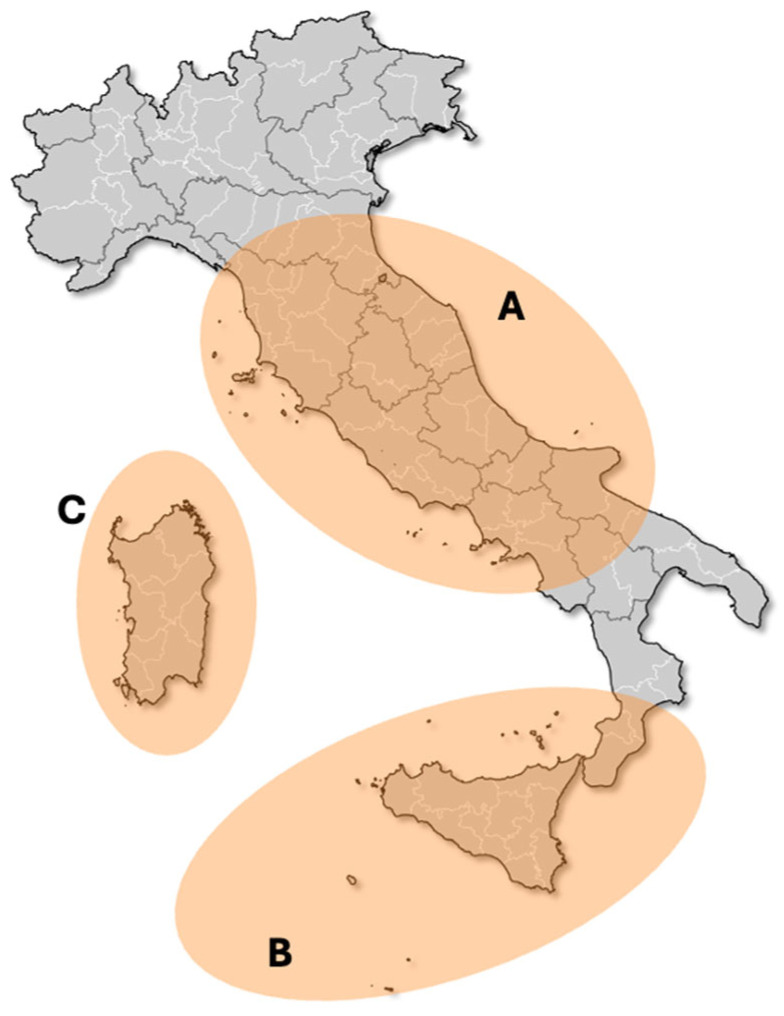
Study areas, namely the three macro-archipelagos discussed in the present study: (**A**) Peninsular Italian archipelago, referred to as circum-Italian; (**B**) Sicily archipelago, referred to as circum-Sicilian; (**C**) Sardinian archipelago, referred to as circum-Sardinian.

**Figure 2 plants-14-01618-f002:**
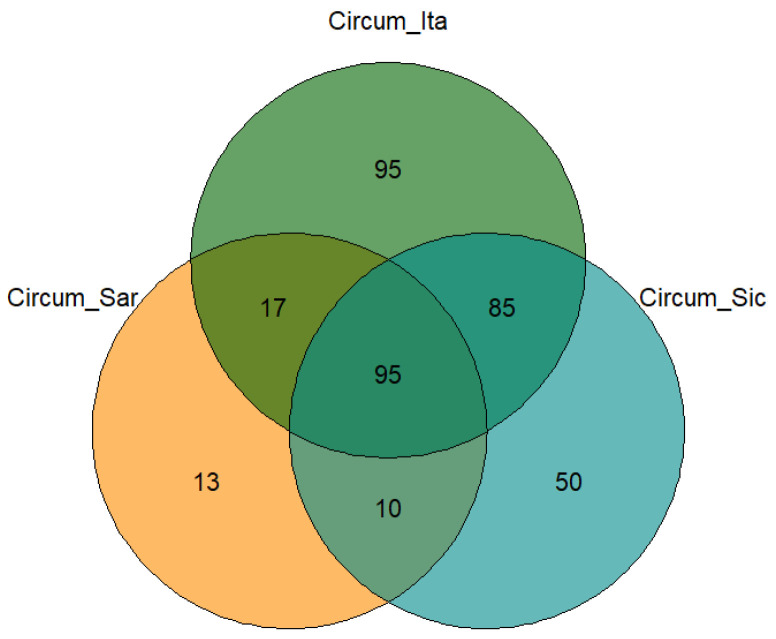
Venn diagram indicating the number of species attributable to each of the studied macro-archipelagos (indicated by different colors and labels).

**Figure 3 plants-14-01618-f003:**
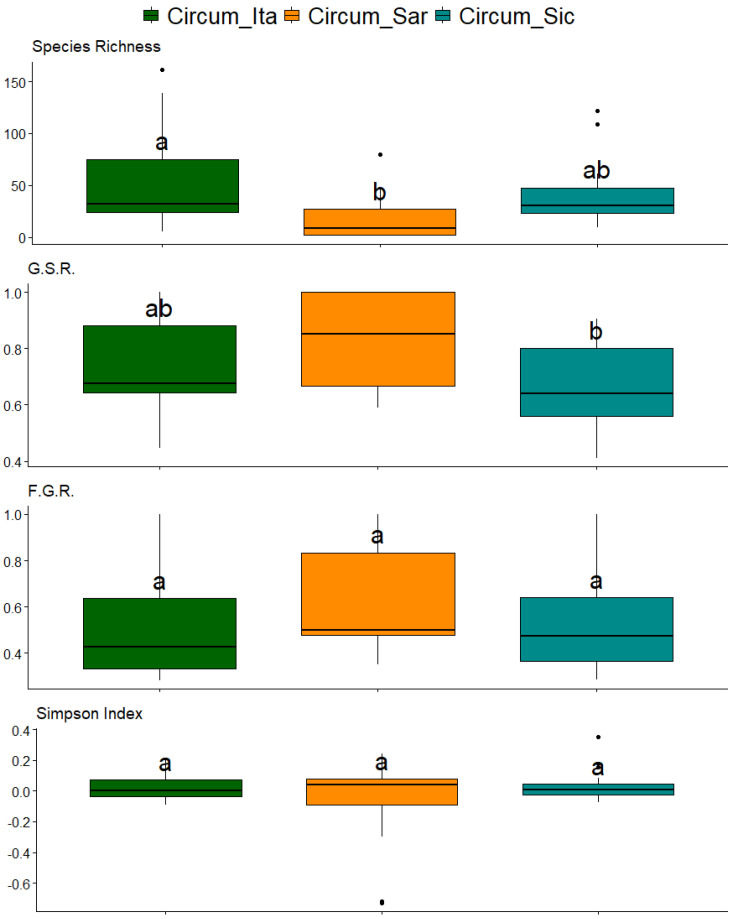
Boxplot representing species richness, G.S.R (genera-to-species ratio), F.G.R. (family-to-genera ratio) and the area-corrected Simpson index of the studied archipelagos. The statistical significance of differences is reported by letters above the boxes. Different letters indicate the statistical significance of differences, while the same letter indicates the absence of any statistically significant difference (differences are considered significant at *p*-value < 0.05). Different archipelagos are indicated by different colors as reported in the legend (Circum_Ita = circum-Italian islands; Circum_Sar = circum-Sardinian islands; Circum_Sic = circum-Sicilian islands).

**Figure 4 plants-14-01618-f004:**
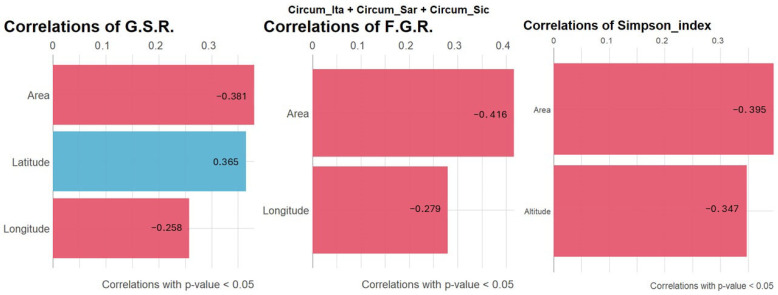
Bar graphs reporting correlation of G.S.R (genera-to-species ratio), F.G.R. (family-to-genera ratio) and of the area-corrected Simpson index with the geographical features considered. Red indicates negative correlations while blue indicates positive correlations. Only significant correlation are reported (*p*-value < 0.05), while the R value is reported on the bar.

**Figure 5 plants-14-01618-f005:**
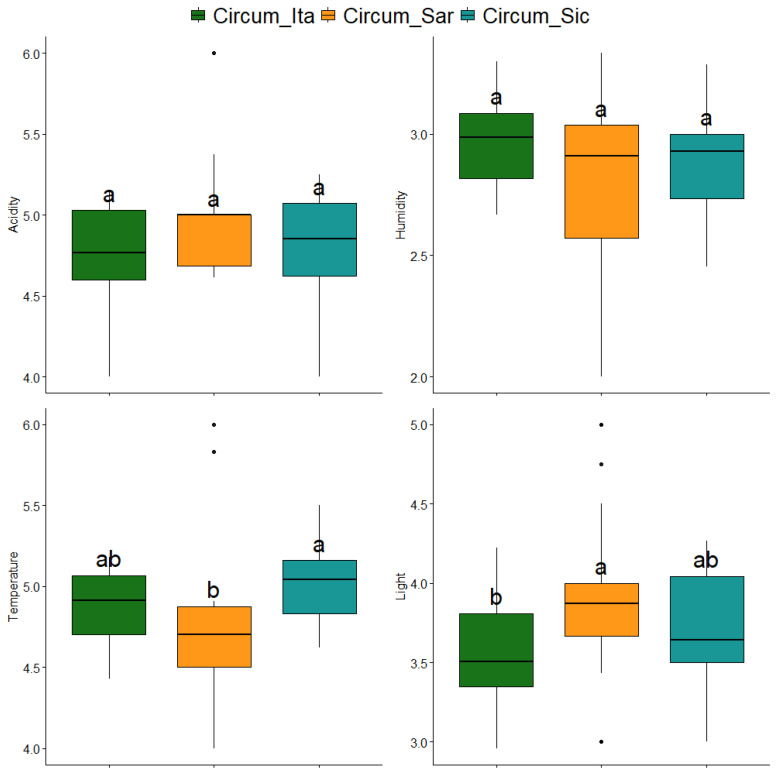
Boxplot representing the mean acidity, humidity, temperature and light requirements in the bryophytes inhabiting the studied archipelagos as described by Dierßen indicators. The statistical significance of differences is reported by letters above the boxes. Different letters indicate the statistical significance of differences, while the same letter indicates the absence of any statistically significant difference (differences are considered significant at *p*-value < 0.05). Different archipelagos are indicated by different colors as reported in the legend (Circum_Ita = circum-Italian islands; Circum_Sar = circum-Sardinian islands; Circum_Sic = circum-Sicilian islands).

**Figure 6 plants-14-01618-f006:**
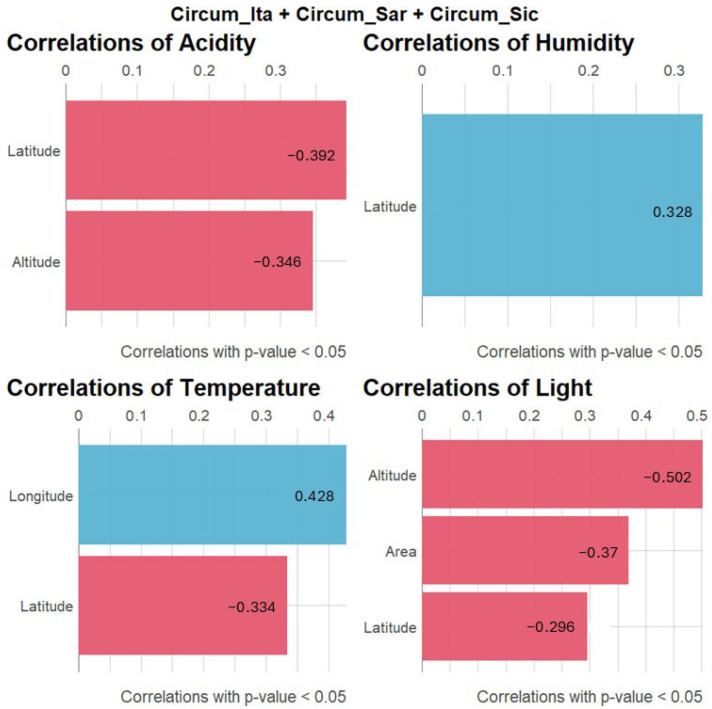
Bar graphics reporting correlation of the Dierßen indicators for the studied species (referring to acidity, humidity, temperature and light requirements) and the geographical features considered. Red indicates negative correlations while blue indicates positive correlations. Only positive correlations are reported (*p*-value < 0.05), while the R value is reported on the bar.

## Data Availability

Data are contained within the article.

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
