# Peer review of "Biogeographical and Ecological Patterns of the Bryophytic Flora Inhabiting the Small Islands Surrounding the Italian Peninsula, Sicily and Sardinia"

_plants, 2025, doi:10.3390/plants14111618_

Round 1
Reviewer 1 Report
Comments and Suggestions for Authors
The topic of the research is quite interesting, and the analyzed material is abundant. Therefore, the publication of the manuscript is justified. However, I see a number of issues that need to be corrected.
At the end of the current version of the introduction:
1) you may add a few sentences about lichens that have similar properties to bryophytes and entering into competitive interactions with the latter
2) it is necessary to formulate and add the goals and objectives of the presented study
Section 3.1.
The authors describe the geology of the islands, but a more important question remains "behind the scenes": were the adjacent islands united with the mainland during, for example, the Pleistocene regressions of the sea? This will say more about the possibilities of settling the islands than even the presence of active volcanism on them (which is undoubtedly also important).
Table 1. It seems to me that it is meaningless. In any table, it is necessary that the information in each cell be determined by some feature not only in the column, but also in the row. In this case, the information in the rows is not structured in any way and cannot be structured. That is, the entire Table 1 is simply a listing of rare species for territorial divisions organized in columns. I believe that the table should be converted into text.
The Discussion section has slipped into notes on the findings of rare species. I would like to see in it a broader theorization answering the questions: how, why and in what way were the floras of the islands surrounding Italy formed; what are the main features of the uniqueness of the island floras in comparison with the Italian mainland;, how is this uniqueness explained and can the insularity effect on the islands surrounding Italy as a whole contribute to the preservation of rare species and evolutionary processes. A simple listing of rare species does little to understand the islands as a refugium of rare species, because rare species can be found in other (almost any) areas of Europe and the globe in general.
The rather wordy conclusion lists the most general facts and "loud phrases" without appropriate references and not following from the text of the article and should be rewritten to make it clear what is the authors' contribution to solving the bright global problems they list.
A number of other comments are in the PDF file.

Author Response
Dear reviewer, Our answer below:
Comments and Suggestions for Authors
The topic of the research is quite interesting, and the analyzed material is abundant. Therefore, the publication of the manuscript is justified. However, I see a number of issues that need to be corrected.
At the end of the current version of the introduction:
1) you may add a few sentences about lichens that have similar properties to bryophytes and entering into competitive interactions with the latter.
A: I thank the reviewer for his comments, and in response to this observation I say that this study only looks at bryophytes and we have not analysed lichens at all.
2) it is necessary to formulate and add the goals and objectives of the presented study.
A: In the revised Word file, the goals and objectives have been addressed.
Section 3.1.
The authors describe the geology of the islands, but a more important question remains "behind the scenes": were the adjacent islands united with the mainland during, for example, the Pleistocene regressions of the sea? This will say more about the possibilities of settling the islands than even the presence of active volcanism on them (which is undoubtedly also important).
A: We agree with the auditor and changes have been incorporated into the text.
Table 1. It seems to me that it is meaningless. In any table, it is necessary that the information in each cell be determined by some feature not only in the column, but also in the row. In this case, the information in the rows is not structured in any way and cannot be structured. That is, the entire Table 1 is simply a listing of rare species for territorial divisions organized in columns. I believe that the table should be converted into text.
A: The Table 1 has had significant changes and enhancements.
The Discussion section has slipped into notes on the findings of rare species. I would like to see in it a broader theorization answering the questions: how, why and in what way were the floras of the islands surrounding Italy formed; what are the main features of the uniqueness of the island floras in comparison with the Italian mainland;, how is this uniqueness explained and can the insularity effect on the islands surrounding Italy as a whole contribute to the preservation of rare species and evolutionary processes. A simple listing of rare species does little to understand the islands as a refugium of rare species, because rare species can be found in other (almost any) areas of Europe and the globe in general.
The rather wordy conclusion lists the most general facts and "loud phrases" without appropriate references and not following from the text of the article and should be rewritten to make it clear what is the authors' contribution to solving the bright global problems they list.
A number of other comments are in the PDF file.
A: In the revised Word file, the comments have been addressed.
We have reloaded the Word file with the corrections.
Best regards
Reviewer 2 Report
Comments and Suggestions for Authors
Generally I find the topic of the manuscript very interesting, as the relationships between bryophyte diversity and island biogeography are little studied and consequently little is known about the possible differences in bryophyte responses as compared to other organisms.
Unfortunately, I found very little specific information trying to decipher these patterns throughour the whole text. Both the Introduction (in particular, li 44-68) and Conclusions sections are very general, without clear ties to the methods applied and results obtained.
As to the methods, the key backgound data for the subsequent analyses must surely be the possibly even sampling effort dedicated to each of the studied islands. This is of course extremely hard to achieve for practical logistical reasons. Therefore, one of the effects which whould be taken care of and clearly presented/admitted, would be specifying details of this effort for individual islands (in relation to their area) and accounting for this effect in the analyses.
Second, I find it very arguable to present the bryophyte diversity of the islands only as the genera-to-species and family-to-genera ratios. These numbers tell us, how much diverse the genera and families are on the average but this has little relevance to the overall bryophyte alfa-diversity in individual islands or their groups. Perhaps, these ratios might be interesting for individual, characteristic genera and families that might serve as a proxy for the overall diversity (such as the genera Riccia, Grimmia or Bryum s.l. perhaps), but not for the whole bryoflora (unless I am missing something, which however should have been specified).
Similarly, the informative value of the shared/exclusive species numbers presented in Fig. 5 is in my opinion arguable, as it is probably largely driven by the total area (or rate of larger islands which are generally richer in species numbers) of islands included in the three considered island groups.
Two of the suggested explications of ecological factors and their effect on diversity also seem to be very arguable: the negative correlation of area and acidity is being explained by the 'dilution of acidic conditions by area size' due to higher biodiversity or different land uses impacting acidity. This looks very cumbersome, is there not simply a chance effect caused by the different rates of occurrence of acidic vs. base-rich substrates (where this rate by chance correlates with the size of analysed islands)? Similarly, the authors find the observed strongly negative reationship between altitude and humidity 'consistent with the well-established concept that higher elevations often exhibit lower humidity levels due to changes in atmospheric pressure and temperature' But I would argue that in general, the opposite is true, the humidity increases with the elevation (except perhaps in some coastal regions at generally lower elevations where the effect of sea might locally prevail).
Formally, the discussional remarks appearing in the Results section should be moved into the DIscussion (which illogically only contains the floristic discussion of rare species which seems to have no relation to the topics analysed elsewhere in the text).
Overall, I find the results to be only publishable if reconsidered in light of the above-mentioned remarks.
Author Response
Dear reviewer, Our answers below:
Comments and Suggestions for Authors
Generally, I find the topic of the manuscript very interesting, as the relationships between bryophyte diversity and island biogeography are little studied and consequently little is known about the possible differences in bryophyte responses as compared to other organisms.
Unfortunately, I found very little specific information trying to decipher these patterns throughout the whole text. Both the Introduction (in particular, li 44-68) and Conclusions sections are very general, without clear ties to the methods applied and results obtained.
R: As to the methods, the key background data for the subsequent analyses must surely be the possibly even sampling effort dedicated to each of the studied islands. This is of course extremely hard to achieve for practical logistical reasons. Therefore, one of the effects which would be taken care of and clearly presented/admitted, would be specifying details of this effort for individual islands (in relation to their area) and accounting for this effect in the analyses.
A: We acknowledge your contribution. To ensure comparable sampling efforts behind species data coming from different studies is indeed crucial, however, in our work this procedure is almost impossible to achieve. We in fact integrate personal sampling campaigns to bibliographical data, some of which lacking any form of specific information about the sampling effort put in place. For this reason, in this new version of the manuscript we chose to exclude data coming from sampling campaigns resulting in very poor species lists (less than 15 individual species per island), since we considered such poor species lists are indicative of an inadequate sampling effort behind that result. At the same time, however, we maintain islands presenting more than 15 records in their species list to avoid the loss of important insights on bryological floras. We specified this in the text.
R: Second, I find it very arguable to present the bryophyte diversity of the islands only as the genera-to-species and family-to-genera ratios. These numbers tell us, how much diverse the genera and families are on the average but this has little relevance to the overall bryophyte alfa-diversity in individual islands or their groups. Perhaps, these ratios might be interesting for individual, characteristic genera and families that might serve as a proxy for the overall diversity (such as the genera Riccia, Grimmia or Bryum s.l. perhaps), but not for the whole bryoflora (unless I am missing something, which however should have been specified).
A: We acknowledge the suggestion, and we added the Shannon index to our study.
R: Similarly, the informative value of the shared/exclusive species numbers presented in Fig. 5 is in my opinion arguable, as it is probably largely driven by the total area (or rate of larger islands which are generally richer in species numbers) of islands included in the three considered island groups.
A: We acknowledge the observation, we modified the text to avoid a non-proper interpretation of the diagram.
R: Two of the suggested explications of ecological factors and their effect on diversity also seem to be very arguable: the negative correlation of area and acidity is being explained by the 'dilution of acidic conditions by area size' due to higher biodiversity or different land uses impacting acidity. This looks very cumbersome, is there not simply a chance effect caused by the different rates of occurrence of acidic vs. base-rich substrates (where this rate by chance correlates with the size of analysed islands)?
A: We modified the manuscript as requested and accepted your explanation to the trend emerging in our study.
R: Similarly, the authors find the observed strongly negative relationship between altitude and humidity 'consistent with the well-established concept that higher elevations often exhibit lower humidity levels due to changes in atmospheric pressure and temperature' But I would argue that in general, the opposite is true, the humidity increases with the elevation (except perhaps in some coastal regions at generally lower elevations where the effect of sea might locally prevail).
A: We re-ran the analysis after pruning data associated with insufficient sampling effort, following your suggestion. As a result, some relationships changed, yielding more ecologically sound results.
Formally, the discussional remarks appearing in the Results section should be moved into the Discussion (which illogically only contains the floristic discussion of rare species which seems to have no relation to the topics analysed elsewhere in the text).
Overall, I find the results to be only publishable if reconsidered in light of the above-mentioned remarks.
A: We have made all the necessary changes, moving most of the results to the discussion section.
We have also changed the conclusions.
We have reloaded the Word file with the corrections.
Best regards
Reviewer 3 Report
Comments and Suggestions for Authors
The study has been well designed, the Introduction and the majority of the Results are presented well. The particular comments on Results and suggestions are shown in the manuscript. Please provide the correlations using Shannon indices as well.
The Discussion has been insufficient to clarify the results and connect them to other published information (it lacks any references!). It also lacks any data on the floristic similarity and dissimilarity between the investigated archipelagos, as well as on the distribution of the rarest species. The Conclusions needs major improvements.
I suggest to add the Supplementary file (table) with the species list and islands they grow on.

I suggest an additional English check.
Author Response
R: The study has been well designed, the Introduction and the majority of the Results are presented well. The particular comments on Results and suggestions are shown in the manuscript. Please provide the correlations using Shannon indices as well.
A: Thank you for the suggestions. We have completely revised the article.
R: The Discussion has been insufficient to clarify the results and connect them to other published information (it lacks any references!). It also lacks any data on the floristic similarity and dissimilarity between the investigated archipelagos, as well as on the distribution of the rarest species. The Conclusions needs major improvements.
A: Thank you for the suggestions. We have completely revised the article.
R: I suggest to add the Supplementary file (table) with the species list and islands they grow on.
A: We have completely revised the article and we add the supplementary table file.
Round 2
Reviewer 2 Report
Comments and Suggestions for Authors
The second version of the manuscript has indeed seen some improvements over the original submission, notably in the Discussion section. However, I still consider the text to suffer from major methodologial flaws and the presented results and conclusions not to be based on real patterns.
The Introduction has not changed much and still mostly contains information not significantly connected with the goal and methods of the text.
In the Methods section, I do not believe that removing islands with <15 known species is an appropriate tool how to discover island with inappropriately low sampling effort. Some of the surveyed islands are very small (< 0.5 km2) and containing few biotopes inhabitable by larger amount of mosses, so in such cases 15 detected bryophytes might be a more relevant number than, say, 40 species from an island of 10 km2 where various habitats occur.
It is not clear how Shannon index was calculated, as I assume that quantitative data on the occurrence of individual species are hardly available for most of the islands, and where there are some data, their representativeness might be arguable.
Another term that is being discussed is insularity but again I am missing any clues how this feature was assessed.
The list of islands in the supplementary data contains all islands included in the previous version.
The conclusions now read like a part of the discussion.
Author Response
Comments and Suggestions for Authors
R: The Introduction has not changed much and still mostly contains information not significantly connected with the goal and methods of the text.
A: We appreciate and understand your suggestion. We have reworded part of the introduction and removed the non-significant parts.
R: In the Methods section, I do not believe that removing islands with <15 known species is an appropriate tool how to discover island with inappropriately low sampling effort. Some of the surveyed islands are very small (< 0.5 km2) and containing few biotopes inhabitable by larger amount of mosses, so in such cases 15 detected bryophytes might be a more relevant number than, say, 40 species from an island of 10 km2 where various habitats occur.
A: We appreciate your comment and in accordance with your suggestion, we reconsidered our approach and opted to retain or exclude islands based on the species-to-area ratio rather than absolute species count. Specifically, we established a threshold ensuring that islands with an exceptionally low density of recorded species were identified as potentially under-sampled rather than simply small in size. This approach is grounded in previous studies and books assessing bryophyte diversity, that highlight the importance of standardizing species richness by area to mitigate the confounding effects of island size and different sampling efforts on biodiversity assessments, e.g., Aleffi et al., 2023, https://doi.org/10.1080/11263504.2023.2284136 and Blasi et al., 2005, Stato della biodiversità in Italia, Palombi editori (book). By applying this criterion, we ensure that islands with proportionally reasonable species richness are retained, while those with unusually low species densities, suggestive of insufficient sampling effort, are excluded from further analyses.
R: It is not clear how Shannon index was calculated, as I assume that quantitative data on the occurrence of individual species are hardly available for most of the islands, and were there are some data, their representativeness might be arguable.
A: We acknowledge the concern regarding the availability and representativeness of the Shannon index in our study case. Apologizing for the lack of clarity, we want to specify that also in the prior version of the manuscript also the Shannon index was calculated based on presence/absence data since, as you correctly pointed out, abundance data in our dataset is absent. However, to address this, we chose to change the index, and did not use the Shannon index but rather the Simpson index, which was calculated based on species presence/absence data rather than abundance data. The Simpson index, although traditionally used with abundance data, can still be calculated from presence/absence data, where it reflects the probability that two randomly selected individuals belong to the same species. In this case, it provides a useful proxy for diversity by accounting for the species evenness, though it will not fully capture the richness as abundance data would. Moreover, in our study, since we found that the Simpson index was significantly influenced by island area (as tested in a linear model). To address this, we corrected for the effect of island size by using the residuals from a regression model with log-transformed island area as the predictor (the log-transformation was applied to normalize the distribution of island sizes and account for potential non-linearity in the relationship between island area and biodiversity). Additionally, as discussed in the prior comment, we excluded islands with very few species records per area to improve data reliability. These adjustments and clarifications are clearly stated in the Materials and Methods section of the manuscript.
R: Another term that is being discussed is insularity but again I am missing any clues how this feature was assessed.
A: The term insularity simply refers to the distance of the island from the nearest mainland (or major island in the case of circum-Sardinian and circum-Sicilian islands and islets) since in English language it indicates the state or condition of being an island or being isolated (both geographically and metaphorically). The higher the distance, the higher the insularity degree of the island. This factor wants to represent the extent to which the bryoflora of each island could be more or less susceptible to the influences of the bryoflora of the mainland, assuming that higher distances represent a stronger barrier in species dispersion. We acknowledge for addressing this lack of clarity in the MS and consequently provided to improve the text.
R: The list of islands in the supplementary data contains all islands included in the previous version.
A: thank you for bringing this error to our attention, we provided to update the tables in the supplementary section. More precisely we updated Table S1 (about the list of islands included in the study and their geographical features) and Table S2 (list of species in the groups of the Venn diagram at Figure 2).
R: The conclusions now read like a part of the discussion.
A: The conclusions and discussion have been separated and modified

Reviewer 3 Report
Comments and Suggestions for Authors
The manuscript has been significantly improved.
However, some issue are still to be resolved:
1. Figures 2, 3, 4, 5 and 6 significantly differ from those in the first version of the manuscript. I would like an explanation why. Did you change anyhow the data used in calculations? The figures are correct indeed, but this discrepancy is very clear to me as a reviewer.
2. Some parts of the manuscript needs further improvements in style and language - this is mostly the case with the chapters Discussion and Conclusion. I strongly recommend the authors to use a native English speaker to improve the language.
3. In the Discussion, many citations of the Figures are present, which should not be the case in the scientific paper. Please, rewrite those sentences using the data and conclusions provided in figures.

I strongly suggest to use native English speaker to improve the language.
Author Response
The manuscript has been significantly improved.
However, some issue are still to be resolved:
R. Figures 2, 3, 4, 5 and 6 significantly differ from those in the first version of the manuscript. I would like an explanation why. Did you change anyhow the data used in calculations? The figures are correct indeed, but this discrepancy is very clear to me as a reviewer.
A. We thank you for your comment, We endeavoured to modify the data in accordance with the recommendations provided by the reviewers. Consequently, the section to which you referred has been revised.
R. Some parts of the manuscript needs further improvements in style and language - this is mostly the case with the chapters Discussion and Conclusion. I strongly recommend the authors to use a native English speaker to improve the language.
A. The complete manuscript has been revised in English.
R. In the Discussion, many citations of the Figures are present, which should not be the case in the scientific paper. Please, rewrite those sentences using the data and conclusions provided in figures.
A. The manuscript was completely revised
Round 3
Reviewer 2 Report
Comments and Suggestions for Authors
I appreciate the amendments made in the last round by the authors, although the new passages unfortunately have not been revised for the corrrectness of English and suffer from many mistakes which must be corrected in the future revision.
You mention volcanism as a critical aspect of the geological history and development of biota, but do not work with the phenomenon elswhere in the text - does this affect the biodiversity as assessed based on the bryofloristic surveys?
The identification of underexplored islands, as the authors made it, now has surely some merit, however, you should be aware that the species-count to area ratio is not linear and you should correct the selection on a more realistic species-to-area ratio.
You state that 'the area-corrected Simpson index decreased 127 in islands presenting higher maximum altitude' but show no difference in Fig. 3 (and, accordingly, state no difference in li111-2)
I do not share the opinion that insularity simply refers to the distance of the island from the nearest mainland. It is a more complex phenomenon, also affected, e.g., by the island size and nature of barriers between the insular and mainland populations (such as the prevailing winds etc.).
It would be useful to add some data to your supplementary table, such as the species count and/or the calculated indices.
It is unclear how the 'most rarely occurring and ecologically important species' were selected to Table 1. Many of them seem to me being rather random occurrences of otherwise more widespread or insufficiently known/recorded[because of defiicult detection or inadequate knowledge] taxa - e.g., Tortula israelis, Trichostomum meridionale, Pseudocrossidium hornschuchianum, Fissidens bryoides var. caespitans, Dicranella schreberiana etc. Others on othe other hand are very specific in requirements, e.g., Fissidens celticus is a pronouncedly oceanic species, so if the record is correct, it must witness markedly oceanic local conditions.
Comments on the Quality of English LanguageThe amendments made to the manuscript in the last round suffer from considerable shortcomings in English (grammar, style and typos) and the language now must be revised.
Author Response
Comments and Suggestions for Authors
R: I appreciate the amendments made in the last round by the authors, although the new passages unfortunately have not been revised for the corrrectness of English and suffer from many mistakes which must be corrected in the future revision.
A: We thank you for your comment, the entire text has been improved in the English language.
R: You mention volcanism as a critical aspect of the geological history and development of biota, but do not work with the phenomenon elswhere in the text - does this affect the biodiversity as assessed based on the bryofloristic surveys?
A: Thank you for your comment. We've gone into more detail about this aspect.
R: The identification of underexplored islands, as the authors made it, now has surely some merit, however, you should be aware that the species-count to area ratio is not linear and you should correct the selection on a more realistic species-to-area ratio.
A: We thank you for your comment regarding the non-linear species-to-area relationship, which is indeed a basic principle in island biogeography. In our study, however, the species-to-area ratio was not employed to model this ecological relationship, but rather as a practical and conservative tool to identify islands that were likely under-sampled. Since our initial dataset was derived from a heterogeneous mix of sources, including personal observations and literature-based species lists compiled over different time periods and under unknown sampling efforts and protocols. So that, traditional approaches that require detailed and standardized sampling data are not applicable. Given these constraints, we opted for a transparent and reproducible filtering method by applying (and clearly stating it in the Materials and Methods section) a threshold species-to-area ratio of 2. This choice was guided by two main goals: to prune out islands with extremely low species counts relative to their area, which may be indicative of under exploration while, at the same time, retaining potentially valuable data from smaller or less surveyed islands without arbitrarily excluding them based on an absolute species count threshold (as we did in the first place). We acknowledge that the S/A ratio does not capture the non-linear character of the species–area relationship, but we believe it provides a reasonable and ecologically intuitive compromise, especially considering the fragmented and uneven nature of the available data in our possess.
We have clarified this rationale in the revised Materials and Methods section and explicitly stated the limitations of our approach.
R: You state that 'the area-corrected Simpson index decreased in islands presenting higher maximum altitude' but show no difference in Fig. 3 (and, accordingly, state no difference in lines 111-2)
A: Thank you for pointing out this incongruence. We provided to better specify in the text that even if, considering the study islands together, no significant difference emerged in the area-corrected Simpson Index, when exploring the correlation between the index and the geographical features of the studied islands, some correlation did appear to be significant (e.g., the negative area-index correlation). This is due to the fact that in the first place we observed the differences in the indexes considering the islands together, while, when exploring the correlations, these were performed considering apart each island to give back a general trend/correlation. We provide an explanation for this in the text.
R: I do not share the opinion that insularity simply refers to the distance of the island from the nearest mainland. It is a more complex phenomenon, also affected, e.g., by the island size and nature of barriers between the insular and mainland populations (such as the prevailing winds etc.).
A: Thank you for your comment. We've gone into more detail about this aspect.
R: It would be useful to add some data to your supplementary table, such as the species count and/or the calculated indices.
A: The changes have been made.
R: It is unclear how the 'most rarely occurring and ecologically important species' were selected to Table 1. Many of them seem to me being rather random occurrences of otherwise more widespread or insufficiently known/recorded [because of defficult detection or inadequate knowledge] taxa - e.g., Tortula israelis, Trichostomum meridionale, Pseudocrossidium hornschuchianum, Fissidens bryoides var. caespitans, Dicranella schreberiana etc. Others on othe other hand are very specific in requirements, e.g., Fissidens celticus is a pronouncedly oceanic species, so if the record is correct, it must witness markedly oceanic local conditions.
A: We removed the table and completely reworked the discussion and the text in general to avoid confusion.

Round 4
Reviewer 2 Report
Comments and Suggestions for Authors
Unfortunately, contrary to authors' statement in the covering letter, the English has not been reviewed and corrected, and not only the cumbersome sentences with incorrect syntax, but even all the typos were transferred to this version (to name a few, e.g., 'surroungin' instead of 'surrounding', li75; 'viewpoing', li76, and many others).
While I understand that it is not easy to apply a scientifically sound criterion for removing the markedly undersampled islands from consideration, it might be better and easy enough to plot the recorded species numbers against the islands' area, and the readings which appear markedly outside the trend line can be discarded as outliers.
I can't see the insularity being explained in more detail. It is only explained that you understand this phenomenon as the distance from the mainland (which in my opinion is not correct, and in this case should simply be named 'distance from mainland'). Supplementary table moreover incorrectly states that this distance is measured in meters (kilometers are correct in my opinion)
Species counts and/or the calculated indices have not been added to the supplementary table, contrary to authors' statement.
The reworked Discussion is now indeed more discussion-like, however, it is also very loose and often without the clear connection to the obtained results, so it is rather an essay on the topic than a discussion proper.
In sum, I can't see a reasonable improvement of this version over the preceding one.
Comments on the Quality of English LanguageAs I can't see a reasonable improvement of this version over the preceding one, I am suggesting the final rejection of the manuscript and would not like to review it anymore. I have spent too much time with this not very interesting text!
Thank you for the understanding, Jan Kucera
Author Response
Unfortunately, contrary to authors' statement in the covering letter, the English has not been reviewed and corrected, and not only the cumbersome sentences with incorrect syntax, but even all the typos were transferred to this version (to name a few, e.g., 'surroungin' instead of 'surrounding', li75; 'viewpoing', li76, and many others).
While I understand that it is not easy to apply a scientifically sound criterion for removing the markedly undersampled islands from consideration, it might be better and easy enough to plot the recorded species numbers against the islands' area, and the readings which appear markedly outside the trend line can be discarded as outliers.
I can't see the insularity being explained in more detail. It is only explained that you understand this phenomenon as the distance from the mainland (which in my opinion is not correct, and in this case should simply be named 'distance from mainland'). Supplementary table moreover incorrectly states that this distance is measured in meters (kilometers are correct in my opinion)
Species counts and/or the calculated indices have not been added to the supplementary table, contrary to authors' statement.
The reworked Discussion is now indeed more discussion-like, however, it is also very loose and often without the clear connection to the obtained results, so it is rather an essay on the topic than a discussion proper.
In sum, I can't see a reasonable improvement of this version over the preceding one.
Answer:
The entire manuscript has undergone a thorough revision in English.
In order to imbue the manuscript with a scientific character, it was necessary to disregard some of the advice given in its previous revision. It is worthy of note that two other reviewers and the publisher accepted the proposed changes.
Best regards,
Silvia Poponessi